# EuclidNets: combining hardware and architecture design for efficient training and inference

## Abstract

In order to deploy deep neural networks on edge devices, compressed (resource efficient) networks need to be developed. While established compression methods, such as quantization, pruning, and architecture search are designed for conventional hardware, further gains are possible if compressed architectures are coupled with novel hardware designs. In this work, we propose EuclidNet, a compressed network designed to be implemented on hardware which replaces multiplication, $wx$, with squared difference $(x - w)^2$. EuclidNet allows for a low precision hardware implementation which is about twice as efficient (in term of logic gate counts) as the comparable conventional hardware, with acceptably small loss of accuracy. Moveover, the network can be trained and quantized using standard methods, without requiring additional training time. Codes and pre-trained models are available at http://github.com/anonymous/.

## 1 Introduction

While the majority of deep neural networks are designed to be implemented on GPUs, they are increasingly being deployed on edge devices, such as mobile phones. These edge devices require compressed (more efficient), *hardware aware* architectures, due to memory and power constraints [7, 11], which seeks to compress the architecture for a given hardware design (e.g. GPU or lower precision chips). However, special-purpose hardware is being designed with neural network inference in mind. This leads to a new problem formulation which we study here: *design an efficient hardware architecture which allows networks to be trained on GPUs, then implemented on the hardware.*

The combined problem of hardware and network design is complex, and the precise measurement of efficiency is both device and problem specific, taking into account latency, memory, energy consumption. Here we deliberately oversimplify the problem in order to make it tractable, by addressing a fundamental element of hardware cost. As a coarse surrogate efficiency, we use the number of logic gates required to implement an arithmetic operation on chip . While this is very coarse, and full costs will depend on other aspects of hardware implementation, it nevertheless represents a fundamental unit of cost in hardware design [23].

In a standard architecture, weights are multiplied by inputs, so the fundamental operation is multiplication $S_{\text{conv}}(x, w) = wx$. In our work, we replace multiplication with the EuclidNet operator,

$$S_{\text{euclid}}(x, w) = -\frac{1}{2}|x - w|^2. \tag{1}$$

which combines a difference with a squaring operator. We will refer to the family of networks that use (1) as EuclidNets. EuclidNets are a compromise between standard architecture, and AdderNets[9], which remove multiplication entirely, but at the cost of a significant loss of accuracy as well as difficulty training. Replacing multiplication with squaring is about half the cost (on chip), depending

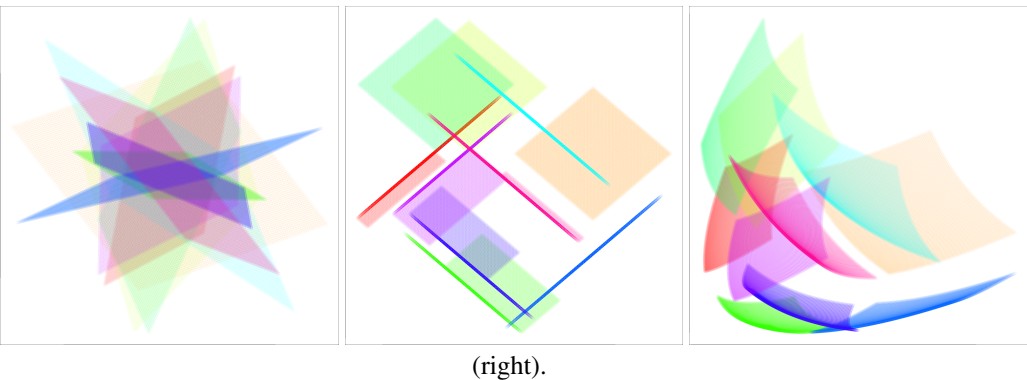

(right).

Figure 1: Feature representation of traditional convolution with $S(x, w) = xw$ (left), AdderNet $S(x, w) = -|x - w|$ (middle), EuclidNet $S(x, w) = -\frac{1}{2}|x - w|^2$

on the number of bits used to represent the integer. The feature representation of each of the architectures is illustrated in Figure 1. EuclidNets can be implemented on 8-bit precision without loss of accuracy, see Table 1.

The squaring operator is cheaper (in terms of logic gates) than multiplication and can be reduced to a tiny look up table if run on integer values. [5, 14] prove replacing look up table can replace actual float computing, but results in practice do not translate to inference speed-up [28]. Works such as LookNN in [38] take the first step in designing hardware for look up table use. On a low precision chip, we can compute $S_{\text{euclid}}$ for about half the cost as $S_{\text{conv}}$, because hardware efficiencies for squaring two a fixed precision integer more than offsets the additional cost of a difference. At the same time, the network does not lose expressivity, as explained below. To summarize, we make the following contributions

- We design an architecture based on replacing the multiplication $S_{\text{conv}}(x, w) = wx$ by the squared difference (1). Quantized networks using this operation require about half the cost (measured by gate operators) on a custom chipset.
- These networks are just as expressive as convolutional networks. In practice, they have comparable accuracy (drop of less than 1 percent on ImageNet on ResNet50 going from full precision convolutional to 8-bit Euclid).
- In contrast to other network compression techniques, we can train and quantize these networks on GPUs without additional cost or difficulty.

Table 1: Euclid-Net Accuracy with full precision and 8-bit quantization: Results on ResNet-20 with Euclidian similarity for CIFAR10 and CIFAR100, and results on ResNet-18 for ImageNet. Euclid-Net achieves comparable or better accuracy with 8-bit precision, compared to the standard full precision convolutional network.

| Network | Quantization | Chip Efficiency | Top-1 accuracy | | |
| | | | CIFAR10 | CIFAR100 | ImageNet |
|---|---|---|---|---|---|
| $S_{\text{conv}}$ | Full precision | ✗ | 92.97 | 68.14 | 69.56 |
| | 8-bit | ✓ | 92.07 | 68.02 | 69.59 |
| $S_{\text{euclid}}$ | Full precision | ✗ | 93.32 | 68.84 | 69.69 |
| | 8-bit | ✓ | 93.30 | 68.78 | 68.59 |
| $S_{\text{adder}}$ | Full precision | ✗ | 91.84 | 67.60 | 67.0 |
| | 8-bit | ✓ | 91.78 | 67.60 | 68.8 |
| BNN | 1-bit | ✓ | 84.87 | 54.14 | 51.2 |

## 2 Context and related work

Neural compression comes at the cost of a loss of accuracy, and may also increase training time (to a greater extent on quantized networks) [19, 12]. Part of the drop in accuracy comes simply from

decreasing model size, which is required for IoT and edge devices [42]. Some of the most common neural compression methods include pruning [39], quantization [21], knowledge distillation [24], and efficient design [27, 25, 47, 41]. Here we focus on a small, unorganized sub-field of compression, that optimizes mathematical operations in the network. This approach can be combined successfully with common other compression methods like quantization [44].

The most natural approach is low bit quantization [21]. The inference gains improves with lowering bit size, at the cost of accuracy drop and longer training. In the extreme case of binary networks, operations have negligible cost at inference but exhibits a considerable accuracy drop [26].

Knowledge distillation [24] consists of transferring information form a larger teacher network to a smaller student network. The idea is easily extended by thinking of information transfer between different similarity measures, which [44] explore in the context of AdderNets. Knowledge distillation is an uncommon training procedure and requires extra implementation effort. EuclidNet keeps the accuracy without knowledge distillation. We suggest a straightforward training using a smooth transition between common convlotution and Euclid operation.

## 3   Network architecture and similarity operators

Consider an intermediate layer of a neural network with input $x \in \mathbb{R}^{H \times W \times c_{\text{in}}}$ and output $y \in \mathbb{R}^{H \times W \times c_{\text{out}}}$ where $H, W$ are the dimensions of the input feature, and $c_{\text{in}}, c_{\text{out}}$ the number of input and output channels, respectively. For a standard convolutional network, represent the transformation from input to output via weights $w \in \mathbb{R}^{d \times d \times c_{\text{in}} \times c_{\text{out}}}$ as

$$y_{mnl} = \sum_{i=m}^{m+d} \sum_{j=n}^{n+d} \sum_{k=0}^{c_{\text{in}}} x_{ijk} w_{ijkl} \tag{2}$$

Setting $d = 1$ recovers the fully-connected layer. We can abstract the multiplication of the weights $w_{ijkl}$ by $x_{ijkl}$ in the equation above by using a similarity measure $S : \mathbb{R} \times \mathbb{R} \to \mathbb{R}$. The convolutional layer corresponds to

$$S_{\text{conv}}(x, w) = xw.$$

In our work, we replace $S_{\text{conv}}$ with $S_{\text{euclid}}$, given by (1). A number of works have also replaced the multiplication operator in a neural network. The most relevant work is the AdderNet of [9], which instead uses

$$S_{\text{adder}}(x, w) = -|x - w|. \tag{3}$$

replacing multiplication by the absolute value of the difference. This operation can be implemented very efficiently on a custom chipset: subtraction and absolute value of a different of $n$-bit integers cost order $n$ gate operations, compared to order $n^2$ for multiplication $S_{\text{conv}}(x, w) = xw$. However, AdderNet comes with a significant loss in accuracy, and is difficult to train.

### 3.1   Other Measures of similarity in neural network architectures

The idea of replacing multiplication operations to save resources within the context of neural networks dates back to 1990s. Equally motivated by computational speed-up and hardware requirement minimization, [17] define perceptrons that use the synapse similarity,

$$S_{\text{synapse}}(x, w) = \text{sign}(x) \cdot \text{sign}(w) \cdot \min(|x|, |w|), \tag{4}$$

which is cheaper than multiplication.

Although (4) has not been experimented with in modern models and datasets, [2] introduced a slight variation, the multiplication-free operator,

$$S_{\text{mfo}}(x, w) = \text{sign}(x) \cdot \text{sign}(w) \cdot (|x| + |w|)). \tag{5}$$

Note that both (4) and (5) induce the $l_1$-norm. [32] explains that the updated design choice allows contributions from both operands $x$ and $w$. [1] studies the similarity in image classification on CIFAR10. Other applications of (5) include [4, 36].

[46] further combines this similarity with a bit-shift, and claims an improved accuracy with negligible added cost. However, the plotted results for AdderNet appear lower than those reported in [9].

Another follow-up work uses knowledge distillation to further improve the accuracy of AdderNets [44].

Instead of simply replacing the similarity on the summation, there is also the possibility to replace the full expression on (2). [30, 31] approximate the activation of a given layer with an exponential term. Unfortunately, it only leads to speed-up in certain cases and, in particular, it does not improve CPU inference time. Reported accuracy on benchmark problems is also lower than the typical baseline.

In a recent work, [34] used three layer morphological neural networks for image classification. Morphological neural networks were introduced in 1990s by [15, 40] and use the notion of erosion and dilation to replace (2):

$$\text{Erosion}(x, w) = \min_j S(x_j, w_j) = \min_j (x_j - w_j),$$

$$\text{Dilation}(x, w) = \max_j S(x_j, w_j) = \max_j (x_j + w_j).$$

The authors propose two methods of stacking layers to expand networks, but admit the possibility of over-fitting and difficult training issues, casting doubt on scalability of the method.

# 4 Theoretical Results for EuclidNets

## 4.1 Expressivity of the EuclidNet network

Networks using the EuclidNet operation as just as expressive as those using multiplication, thanks to the polarization identity,

$$S_{\text{conv}}(x, w) = S_{\text{euclid}}(x, w) - S_{\text{euclid}}(x, 0) - S_{\text{euclid}}(0, w)$$

which means that any multiplication operation can be expressed using only Euclid operations.

## 4.2 Logic Gate Cost for EuclidNet compared to ConvNet (multipication)

The above similarity may not come across immediately as an improved choice on the cost of convolutions. It requires personalized hardware to obtain gains in inference speed like the other similarities. For example, in a typical architecture, the cost of addition is very close to multiplication, and squaring is usually not considered distinctly from multiplication [30, Table III]. Hence, first we discuss what these gains are theoretically. As for training, unlike other competitors such as AdderNet that embodies a considerable slow training, we implement the Euclid similarity in a way that is only slightly slower than $S_{\text{conv}}$.

Here we provide a brief theoretical analysis of basic binary operations on custom hardware that is optimized for model inference. Assuming equal cost between AND, XOR and OR gates, we first compute the cost of gate-level integer operations, defined in Appendix A.1. See Figure 2

The following formula gives the gate count of $n$-bit operations:

$$S_{\text{conv}} = 6n^2 - 8n + 3$$

$$S_{\text{euclid}} = 3n^2 + n/2 - 3$$

(with a minor modification to the second formula to $3n^2 + n/2 - 3/2$ when $n$ is odd), refer to Table A.4.

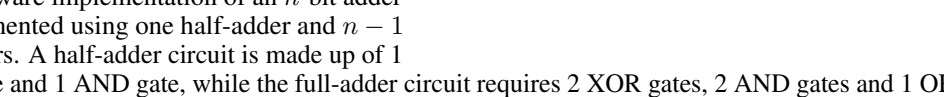

Figure 2: Comparison of the number of logic gates ($y$-axis) as a function of the number of bits ($x$-axis) EuclidNet compared with the standard ConvNet.

The hardware implementation of an $n$-bit adder is implemented using one half-adder and $n - 1$ full-adders. A half-adder circuit is made up of 1 XOR gate and 1 AND gate, while the full-adder circuit requires 2 XOR gates, 2 AND gates and 1 OR gate. Therefore, the cost of an $n$ bit addition is $5n - 3$.

Table 2: Time (seconds) and maximum training batch-size that can fit in a signle GPU *Tesla V100-SXM2-32GB*, during ImageNet training. In parenthesis is the slowdown with respect to the $S_{conv}$ baseline. We do not show times for AdderNet, which is much slower than both, because it is not implemented in CUDA

| Model | Method | Maximum Batch-size | | Time per step | |
| | | power of 2 | integer | Training | Testing |
| --- | --- | --- | --- | --- | --- |
| ResNet-18 | $S_{\text{conv}}$ | 1024 | 1439 | 0.149 | 0.066 |
| | $S_{\text{euclid}}$ | 512 | 869 (1.7×) | 0.157 (1.1×) | 0.133 (2×) |
| ResNet-50 | $S_{\text{conv}}$ | 256 | 371 | 0.182 | 0.145 |
| | $S_{\text{euclid}}$ | 128 | 248 (1.5×) | 0.274 (1.5×) | 0.160 (1.1×) |

There are $n^2$ AND gates for $n$-bit element wise multiplications. A common architecture usually include $(n - 1)$ $n$-bit adders besides the $n^2$ AND gates. One $n$-bit adders is composed of one half-adder and $n - 1$ full-adders. Hence the cost of multiplication is $6n^2 - 8n + 3$.

In the case of squaring, there are less AND gates representing element-wise multiplication. We consider two different cases: i) if $n$ is **even** the cost of squaring is $3n^2 - \frac{9}{2}n$ ii) if $n$ is **odd**, the cost of squaring is $3n^2 - \frac{9}{2}n + \frac{3}{2}$,

## 5   Training EuclidNets

Training EuclidNets are much easier compared with other competitors such as AdderNets. This makes EuclidNet attractive for complex tasks such as image segmentation, and object detection where training compressed networks are challenging and causes large accuracy drop. However, EuclidNets are more expensive than AdderNets on floating points, but their quantization behavior unlike AdderNets resembles traditional convolution to a great extent. In another words EuclidNets are easiy to quantize.

While training a network, it is more appropriate to use the identity

$$S_{\text{euclid}}(x, w) = -\frac{x^2}{2} - \frac{w^2}{2} + xw,$$ (6)

and use this equation while training EuclidNets on GPUs which are optimized for inner product. Therefore training EuclidNets doesn't require additional CUDA core [35] implementation unlike AdderNets. The official implementation of AdderNet [9] reflects order of $20\times$ slower training than the traditional convolution on Pytorch. This is specially problematic for large networks and complex tasks that even traditional convolution training takes few days or even weeks. EuclidNet training is $2\times$ in the worst case and their implementation is natural in deep learning frameworks such as PyTorch and Tensorflow.

A common method in training neural networks is fine-tuning, initializing with weights trained on different data but with a similar nature. Here, we introduce the idea of using a weight initialization from a model trained on a related similarity.

Rather than training from scratch, we wish to fine-tune EuclidNet starting from accurate CNN weights. This is achieved by an "architecture homotopy" where we change hyperparameters to convert a regular convolution to an Euclid operation

$$S(x, w; \lambda_k) = xw - \lambda_k \frac{x^2 + w^2}{2}, \qquad \text{with } \lambda_k = \lambda_0 + \frac{1 - \lambda_0}{n} \cdot k,$$ (7)

where $n$ is the total number of epochs and $0 < \lambda_0 < 1$ is the initial transition phase. Note that $S(x, w, 0) = S_{\text{conv}}(x, w)$ and $S(x, w, 1) = S_{\text{euclid}}(x, w)$ and equation 7 is the convex combination of the two similarities. One may interpret $\lambda_k$ as a schedule for the homotopy parameter, similar to how a schedule is defined for the learning rate in training a deep network. We found that a linear schedule above is effective empirically.

Transformations like (7) are commonly used in scientific computing [3]. The idea of using homotopy in training neural networks can be traced back to [13]. Recently, homotopy was used in deep learning in the context of activation functions [37, 8, 33, 18], loss functions [20], compression [10] and transfer learning [6]. Here, we use homotopy in the context of transforming network operations.

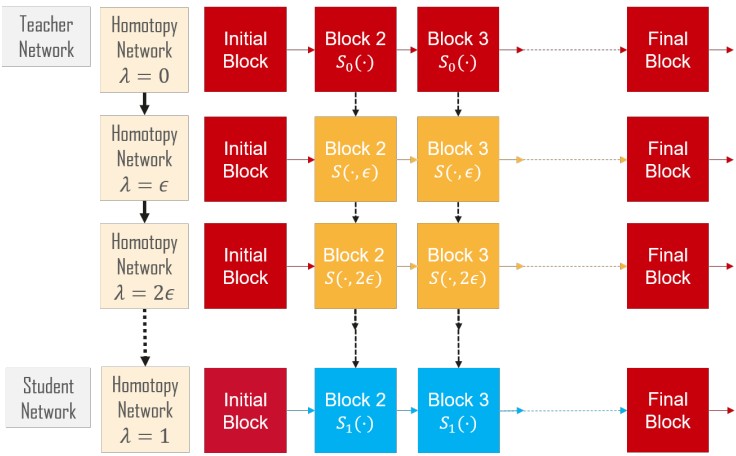

Figure 3: Training schema of EuclidNet using Homotopy, i.e. transitioning from traditional convolution $S(x, w) = xw$ towards EuclidNet $S(x, w) = -\frac{1}{2}|x - w|^2$ through equation (7).

Fine-tuning method in (7) is inspired by continuation methods in partial differential equations. Assume $S$ is a solution for a differential equation with the initial condition $S(x, 0) = S_0(x)$. In certain situations, solving this differential equation for $S(x, t)$ and then evaluating at $t = 1$ might be simpler than solving directly for $S_1$. One may think of this homotopy method as an evolving neural network over time. At time zero the neural network consists of regular convolutional layers, but at time one transforms to Euclidean layers.

The homotopy method can be interpreted as a sort of of knowledge distillation. Whereas knowledge distillation methods tries to match a student network to a teacher network, the homotopy can be seen as a slow transformation from the teacher network into a student network. Figure 3 shows a scheme of the idea. Curiously, problems that have been solved with homotopic approaches have also been tackled by knowledge distillation. For example, removing blocks or layers from a network [24, 10] along with transfer learning [45, 6].

# 6 Experiments

We consider try our proposed method on image classification task. Future work could be extended to other domains of application such as natural language and speech.

## 6.1 CIFAR10

First, we consider the CIFAR10 dataset, consisting of $32 \times 32$ RGB images with 10 possible classifications [29]. We normalize and augment the dataset with random crop and random horizontal flip. We consider two ResNet models [22], ResNet-20 and ResNet-32.

We train EuclidNet using the optimizer from [9], which we will refer to as AdderSGD, to evaluate EuclidNet under a similar setup. We use initial learning rate $0.1$ with cosine decay, momentum $0.9$ and weight decay $5 \times 10^{-4}$. We follow [9] in setting the learning-rate scaling parameter $\eta$. However, we use a batch-size of 128 for memory reasons. For traditional convlution network, we use the same hyper-parameters with stochastic gradient descent optimizer.

In Table 3 we provide the details of classification accuracy. We consider two different weight initialization for EuclidNets. First, we initialize randomly and second, we initialize from weights pre-trained on a convolutional network. The accuracy for EuclidNets is approximately the same as for a standard ResNet. We see that for CIFAR10 training from scratch achieves even a higher accuracy, while initializing with convolution network and using linear Homotopy training improves it even further.

During training, EuclidNets are unstable, despite careful choice of the optimizer. In Figure 4 we compare with training the corresponding convolutional network. Fine-tuning directly from

Table 3: Results on CIFAR10. The initial learning rate is adjusted for non-random initialization.

| Model | Similarity | Initialization | Homotopy | Epochs | Top-1 accuracy CIFAR10 | Top-1 accuracy CIFAR100 |
|---|---|---|---|---|---|---|
| ResNet-20 | $S_{\text{conv}}$ | Random | None | 400 | 92.97 | **69.29** |
| | $S_{\text{euclid}}$ | Random | None | 450 | 93.00 | 68.84 |
| | | Conv | None | 100 | 90.45 | 64.62 |
| | | | Linear | 100 | **93.32** | 68.84 |
| ResNet-32 | $S_{\text{conv}}$ | Random | None | 400 | **93.93** | 71.07 |
| | $S_{\text{euclid}}$ | Random | None | 450 | 93.28 | **71.22** |
| | | Conv | None | 150 | 91.28 | 66.58 |
| | | | Linear | 100 | 92.62 | 68.42 |

Table 4: Full precision results on ResNet-20 for CIFAR10 for different multiplication-free similarities.

| Similarity | $S_{\text{conv}}$ | $S_{\text{euclid}}$ | $S_{\text{adder}}$ | $S_{\text{mfo}}$ | $S_{\text{synapse}}$ |
|---|---|---|---|---|---|
| **Accuracy** | 92.97 | **93.00** | 91.84 | 82.05 | 73.08 |

convolutional weights is more stable than training from scratch as expected. However, accuracy is lower but the convergence is faster when we use homotopy training and the accuracy is improved. Pre-trained convolution weights are commonly available in the most of neural compression tasks, so initializing EuclidNets with pre-trained convolution is more natural and preferable.

EuclidNets are not only faster to train compared with other competitors, but also stand superior in terms of accuracy. AdderNet performs slightly worse but is much slower to train. The accuracy is significantly lower for the synapse and the multiplication-free operator. In Table 4 we record top-1 accuracy obtained in which Adder-Net results are borrowed from [44], that use knowledge distillation to close the gap with the full precision but still falls short compared with EuclidNet.

Training a quantized $S_{\text{euclid}}$ is very similar similar to convolution. This allows a wider use of such networks for lower resource devices. Quantization of the Euclid model to 8bits keeps accuracy drop within the range of one percent [43] similar to traditional convolution so they are like convolution when run on lower bits. Table 1 shows 8-bit quantization of EuclidNet where the accuracy drop remains negligible. Similar to traditional convolution, EuclidNets on CIFAR100 exhibit a larger accuracy drop compared to CIFAR10, probably due to the complexity of the classification problem.

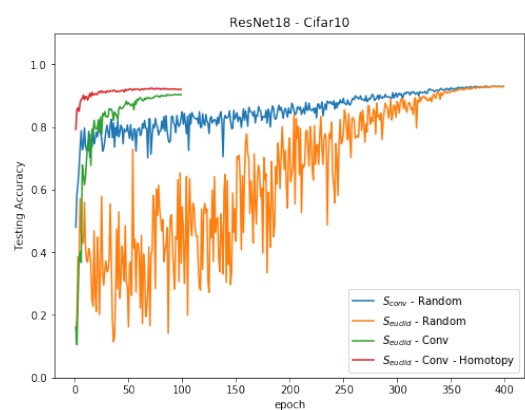

Figure 4: Evolution of testing accuracy during training of ResNet-20 on CIFAR10, initialized with random weights, or initialized from convolution pre-trained network. Initializing from a pre-trained convolution network speeds up the convergence. EuclidNet is harder to train compared with convolution network when both initialized from random weights.

## 6.2 ImageNet

Next, we consider EuclidNet classifier built on ImageNet, a more challenging task ImageNet [16]. We train our baseline with standard augmentations of random resized crop and horizontal flip and normalization. We consider ResNet-18 and ResNet-50 models. Hyper-parameters tuning follows Section 6.1.

Table 5 shows top-1 and top-5 classification accuracies. The accuracy from while EuclidNet is trained from scratch is lower, showing the importance of homotopy training. We believe that the accuracy drop with no homotopy is the difficulty of tuning training hyper-parameters for a large dataset such as ImageNet. Even though hyper-parameters that achieve equivalent accuracy from random initialization

exist, they are too difficult to find. It is much easier to use the existing hyperparameters of traditional convolution, and transfer the geometry through homotopy training.

Table 5: Full precision results on ImageNet. Best result for each model is in bold.

| Model | Similarity | Initialization | Homotopy | Epochs | Top-1 Accuracy | Top-5 Accuracy |
|-------|-----------|----------------|----------|--------|----------------|----------------|
| | $S_{conv}$ | Random | None | 90 | 69.56 | 89.09 |
| | | Random | None | 90 | 64.93 | 86.46 |
| ResNet-18 | $S_{euclid}$ | | None | 90 | 68.52 | 88.79 |
| | | Conv | | 10 | 65.36 | 86.71 |
| | | | Linear | 60 | 69.21 | 89.13 |
| | | | | 90 | **69.69** | **89.38** |
| | $S_{conv}$ | Random | None | 90 | 75.49 | 92.51 |
| | | Random | None | 90 | 37.89 | 63.99 |
| ResNet-50 | $S_{euclid}$ | | None | 90 | 75.12 | 92.50 |
| | | Conv | | 10 | 70.66 | 90.10 |
| | | | Linear | 60 | 74.93 | 92.52 |
| | | | | 90 | **75.64** | **92.86** |

## 7 Conclusion

Euclid networks are obtained from typical neural models by replacing multiplication in convolutional layers by the Euclidean similarity. They are designed to be implemented on a custom designed low precision chipset, with the idea that subtraction and squaring can be implemented using approximately half the logic gates, compared to multiplication.

While other efficient architectures can be difficult to train in low precision, EuclideNets are easily trained in low precisions. EuclidNets can be initialized with weights trained on the correspondent ConvNet to save training time, so on may regard them as a fine tuning convolutiuonal networks for a cheaper inference. The homotopy method further improves training in such scenarios and training using this method sometimes surpass regular convolution accuracy. Future work may focus on developing hardware that can realize the expected inference time losses and try similar experiments on down stream vision tasks like object detection and segmentation.

### 7.1 Limitations

While gate counts provide a fundamental method for assessing the cost of a chip, they are a crude estimate, and the real costs (in terms of power usage, inference time, and memory) of a chipset and architecture combination are much more complex to estimate. True final costs can require a hardware simulator or implementation. At the same time, the gate count provides a first approximation to the cost, and the fact that we can train and match accuracy in eight bit precision is promising.

### 7.2 Societal Impact

Deep Neural Network inference is costly in terms of power usage. If we can design and implement efficient architectures, this will reduce the societal cost of running these models on edge devices.

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
