## A  Appendix

### A.1  Hardware Details

We compute the number of logic gates required for each integer operation.

### A.2  Addition

A half-adder (HA) circuit is made up of 1 XOR gate and 1 AND gate, while the full-adder (FA) circuit requires 2 XOR gates, 2 AND gates and 1 OR gate. Therefore, the cost of an $n$ bit addition is

$$
\begin{aligned}
&\text{HA} + (n-1) \times \text{FA} \\
&= (1\,\text{XOR} + 1\,\text{AND}) + (n-1) \times (2\,\text{XOR} + 2\,\text{AND} + 1\,\text{OR}) \\
&= (2n-1)\,\text{AND} + (2n-1)\,\text{XOR} + (n-1)\,\text{OR} \\
&\approx 5n - 3
\end{aligned}
$$

### A.3  Multiplication

A common architecture usually include $(n-1)$ $n$-bit Adders besides the $n^2$ AND gates, see Figure 5 top panels. One $n$-bit adders is composed of one half-adder (HA) and $n-1$ full-adder (FA). We will consider a $n$-bit adder as building block in our theoretical analysis, although it could be optimized further.

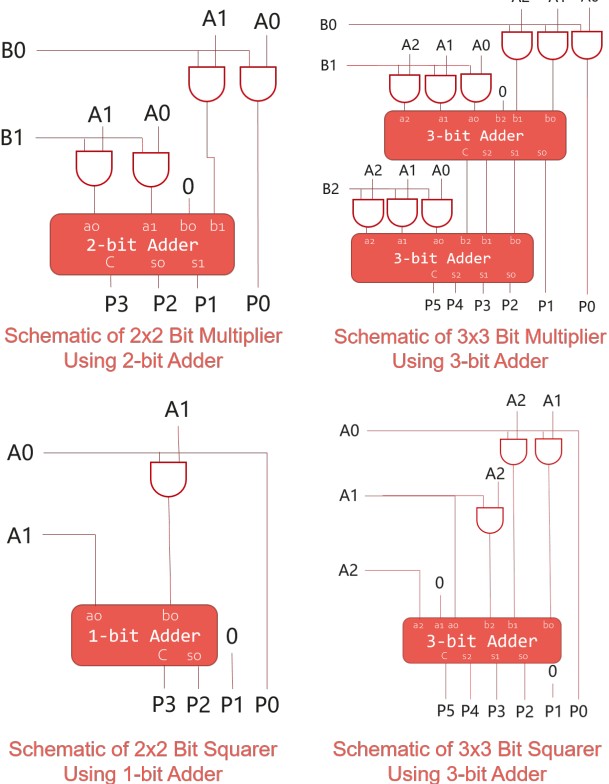

Figure 5: Binary multiplier (top panel) and binary squarer (bottom panels) for number of bits $n = 2$ (left panels) and $n = 3$ (right panels).

Hence the cost of multiplication is

$$n^2 \text{ AND} + (n-1) \times (n-\text{bit Adder})$$
$$= n^2 \text{ AND} + (n-1) \times \text{HA} + (n-1)^2 \times \text{FA}$$
$$= n^2 \text{ AND} + (n-1) \times (1 \text{ XOR} + 1 \text{ AND}) + (n-1)^2 \times (2 \text{ XOR} + 2 \text{ AND} + 1 \text{ OR})$$
$$= (3n^2 - 3n + 1) \text{ AND} + (2n^2 - 3n + 1) \text{ XOR} + (n^2 - 2n + 1) \text{ OR}$$
$$\approx 6n^2 - 8n + 3$$

## A.4 Squaring

In the case of squaring, we have less AND gates representing element-wise multiplication, because some values are repeated. We provide some examples in Figures 6 and 7.

| | | | $A_3$ | $A_2$ | $A_1$ | $A_0$ |
|---|---|---|---|---|---|---|
| | | | $A_3$ | $A_2$ | $A_1$ | $A_0$ |
| | | | $A_0A_3$ | $A_0A_2$ | $A_0A_1$ | $A_0^2$ |
| | | $A_1A_3$ | $A_1A_2$ | $A_1^2$ | $A_1A_0$ | 0 |
| | $A_2A_3$ | $A_2^2$ | $A_2A_1$ | $A_2A_0$ | 0 | 0 |
| $A_3^2$ | $A_3A_2$ | $A_3A_1$ | $A_3A_0$ | 0 | 0 | 0 |
| $A_3^2$ | $2(A_2A_3)$ | $A_2^2 + 2(A_1A_3)$ | $2(A_0A_3) + 2(A_1A_2)$ | $A_1^2 + 2(A_0A_2)$ | $2(A_0A_1)$ | $A_0^2$ |

Figure 6: Binary Square for $n = 4$ bits.

| | | | | $A_4$ | $A_3$ | $A_2$ | $A_1$ | $A_0$ |
|---|---|---|---|---|---|---|---|---|
| | | | | $A_4$ | $A_3$ | $A_2$ | $A_1$ | $A_0$ |
| | | | | $A_0A_4$ | $A_0A_3$ | $A_0A_2$ | $A_0A_1$ | $A_0^2$ |
| | | | $A_1A_4$ | $A_1A_3$ | $A_1A_2$ | $A_1^2$ | $A_1A_0$ | 0 |
| | | $A_2A_4$ | $A_2A_3$ | $A_2^2$ | $A_2A_1$ | $A_2A_0$ | 0 | 0 |
| | $A_3A_4$ | $A_3^2$ | $A_3A_2$ | $A_3A_1$ | $A_3A_0$ | 0 | 0 | 0 |
| $A_4^2$ | $A_4A_3$ | $A_4A_2$ | $A_4A_1$ | $A_4A_0$ | 0 | 0 | 0 | 0 |
| $A_4^2$ | $2(A_3A_4)$ | $A_3^2 + 2(A_2A_4)$ | $2(A_1A_4 + A_2A_3)$ | $A_2^2 + 2(A_0A_4 + A_1A_3)$ | $2(A_0A_3 + A_1A_2)$ | $A_1^2 + 2(A_0A_2)$ | $2(A_0A_1)$ | $A_0^2$ |

Figure 7: Binary Square for $n = 5$ bits.

In Figures 6 and 7, we see that some sums are actually a multiplication by a factor of 2. Multiplication by a factor of 2 can instead be though as a shift towards the left in the addition.

1. If $n$ is **even**, then only the middle column will shift $\lfloor \frac{n}{2} \rfloor = \frac{n}{2}$ values to the left. Also, the column on the left will have the term $A_{n-1}^2$. So, the sum with maximum number of elements, $\frac{n}{2} + 1$, will only happen in one column, $i = n - 1$. Hence, we need $\frac{n}{2} (n-1)$-bit adders. See Figure 8 for visual intuition.

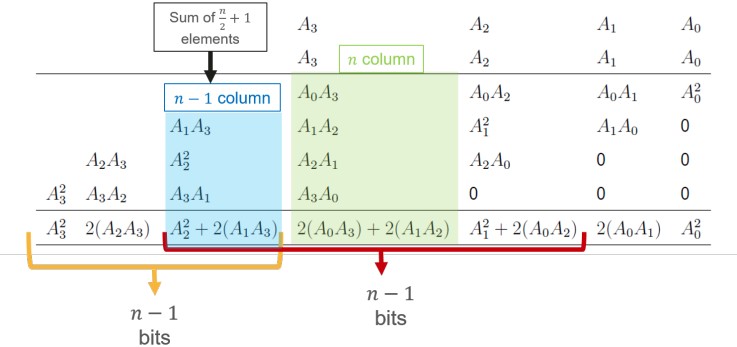

Figure 8: Intuition for square on $n$ even.

Hence, the cost of squaring when $n$ is even is:

$$\frac{n(n-1)}{2} \text{ AND} + \frac{n}{2} \times ((n-1) - \text{bit Adder})$$

$$= \frac{n(n-1)}{2} \text{ AND} + \frac{n}{2} \times \text{HA} + \frac{n}{2}(n-2) \times \text{FA}$$

$$= \frac{n(n-1)}{2} \text{ AND} + \frac{n}{2} \times (1 \text{ XOR} + 1 \text{ AND}) + \frac{n}{2}(n-2) \times (2 \text{ XOR} + 2 \text{ AND} + 1 \text{ OR})$$

$$= \left(\frac{3}{2}n^2 - 2n\right) \text{ AND} + \left(n^2 - \frac{3}{2}n\right) \text{ XOR} + \left(\frac{1}{2}n^2 - n\right) \text{ OR}$$

$$\approx 3n^2 - \frac{9}{2}n$$

2. If $n$ is **odd**, column $i = n - 1, n, n + 1$ will shift $\lfloor\frac{n}{2}\rfloor = \frac{n-1}{2}$ values to the left. Since columns $i = n - 2, n$ both have an $A_i^2$ term, the sum with maximum number of elements, $\frac{n-1}{2} + 1$, will happen at those columns. Hence, we need $\frac{n-1}{2}$ $n$-bit adders. See Figure 9 for visual intuition.

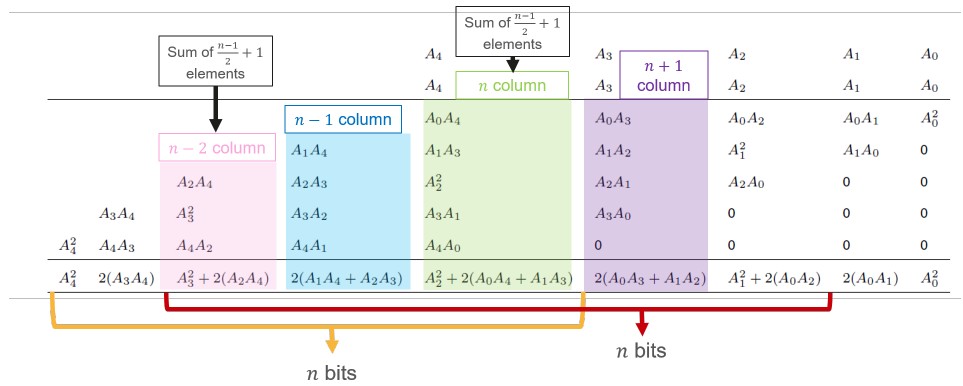

Figure 9: Intuition for square on $n$ odd.

| Similarity | | Gate Count |
|---|---|---|
| $S_{\text{conv}}$ | | $6n^2 - 8n + 3$ |
| $S_{\text{euclid}}$ | $n$ odd | $3n^2 + \frac{1}{2}n - \frac{3}{2}$ |
| | $n$ even | $3n^2 + \frac{1}{2}n - 3$ |

Figure 10: Similarity operator Gate Count.

Hence, the cost of squaring when $n$ is odd is:

$$\frac{n(n-1)}{2} \text{ AND} + \frac{n-1}{2} \times (n - \text{bit Adder})$$
$$= \frac{n(n-1)}{2} \text{ AND} + \frac{n-1}{2} \times \text{HA} + \frac{n-1}{2}(n-1) \times \text{FA}$$
$$= \frac{n(n-1)}{2} \text{ AND} + \frac{n-1}{2} \times (1 \text{ XOR} + 1 \text{ AND}) + \frac{n-1}{2}(n-1) \times (2 \text{ XOR} + 2 \text{ AND} + 1 \text{ OR})$$
$$= \left(\frac{3}{2}n^2 - 2n + \frac{1}{2}\right) \text{ AND} + \left(n^2 - \frac{3}{2}n + \frac{1}{2}\right) \text{ XOR} + \left(\frac{1}{2}n^2 - n + \frac{1}{2}\right) \text{ OR}$$
$$\approx 3n^2 - \frac{9}{2}n + \frac{3}{2}.$$

Moreover, in Figure 5 (bottom panels), we present the corresponding hardware schemes for $n = 2, 3$.

| Operation | | Gate Count |
|---|---|---|
| Add | | $5n - 3$ |
| Multiply | | $6n^2 - 8n + 3$ |
| Square | $n$ odd | $3n^2 - \frac{9}{2}n + \frac{3}{2}$ |
| | $n$ even | $3n^2 - \frac{9}{2}n$ |

| Similarity | | Gate Count |
|---|---|---|
| $S_{\text{conv}}$ | | $6n^2 - 8n + 3$ |
| $S_{\text{euclid}}$ | $n$ odd | $3n^2 + \frac{1}{2}n - \frac{3}{2}$ |
| | $n$ even | $3n^2 + \frac{1}{2}n - 3$ |

Figure 11: Logic gate count for operations $n$-bit integers.