# OpenReview forum: "EuclidNets: Combining hardware and architecture design for Efficient Inference and Training"
_NeurIPS.cc/2021/Conference — NeurIPS 2021 Submitted_

### Official Review · Reviewer_TQtz · 2021-07-16

**Rating:** 7
**Confidence:** 4

**Summary:**

==+== A. Paper summary ==+==

This paper presents EuclidNet, a method to replace multiplication in traditional ConvNet with l2 or Euclid distance. Results show that fine-tuning from a pretrained CNN to a EuclidNet can achieve similar or better results compared to traditional ConvNet and Addernet.

**Ethical Concerns:**

No ethical concerns

**Limitations And Societal Impact:**

Have the authors adequately addressed the limitations and potential negative societal impact of their work? Yes.



**Main Review:**

==+== B. Strengths

-- The idea of the paper is good and thought-provoking. Equation 4.1 indicates that EulidNet lies in the middle between quantized and full-precision models.

-- Result shows the accuracy/fine-tuning cost is low.

==+== C. Weakness

--- The presentation can be improved a lot.

--  The evaluation is not thorough.

--  Training from scratch is unstable.

Questions:

Do we need two squaring hardware to handle odd/even numbers? Figure 2 is confusing. And if we double the size of logic gates for squaring, it would show larger cost than multiplication.

I can see one downside of EulidNet is that it cannot exploit sparsity like traditional ConvNet. In my opinion, your work is orthogonal to other works dealing with model sparsity; and you should compare your method with them.

Figure 3 is confusing, are you searching the best $\lambda$ for fine-tuning using linear grid search? If that is the case, wouldn’t this greatly increase your fine-tuning time?

The evaluation section is also confusing and a lot of details are missing. For example, I don’t know if Table 3 is for quantized models or full-precision models. If it is for a full-precision model, I suggest you add a quantization baseline to show your method is better than directly quantizing a ConvNet.

Regarding the hardware cost, I suggest the author find an efficient hardware implementation of squaring. You can synthesize a piece of hardware (FP squaring v.s. FP multiplication) to see the simulated runtime cycles. As traditional metric (FLOPs) is not working in your scenario, it is also better to come up with a new metric (e.g., cycles) to evaluate your runtime cost against other baselines.

**Time Spent Reviewing:**

2

---

> ### Author Response · Authors · 2021-08-09
> **Full response**
>
> Thank you for taking the time to review our paper. It appears you have captured the main point of our paper which was improving efficiency while retaining accuracy close to convolutional neural networks. Also, thank you for recognizing the paper would be a contribution to the literature.
>
> However, you express valid concerns we would like to address. We believe points 1 and 3 could easily be addressed in a revision to the paper, while point 2 may be more appropriate for follow-up work. We go into detail next:
>
> 1. Regarding the missing experimental setting in Table 1, there are two different things to clarify: 1) we can easily add references to the rows of S_adder and BNN as the values were copied from references [9] and [44] already cited. We may also add a short paragraph describing the experimental setting as they appear in this reference. As for the remaining rows, we can add a sentence referring the reader to Section 6 where the experimental details are described. To address your more general issue of paper organization, the reason we chose to display Table 1 at the beginning is so our contributions and impact are clear from the start without overwhelming the reader with details. We still have more in depth justifications, but in the following sections of the paper. We hope being more careful in guiding the reader to those sections helps with your concern.
> 2. This is a great suggestion, but also a major additional step. Our understanding is that new architectures are already a major big step with demonstration of training and accuracy. The reason we provided an intuitive theoretical gate estimation is already to improve upon other literature papers that do not consider hardware (AdderNet works for example [9,44]). We may not have implemented hardware with the expertise of ShiftAddNet [46], because we focused instead in speeding up training by developing the homotopy method, which is not usually addressed in other papers. Additionally, only recently has a hardware implementation been done for AdderNet [*].
> 3. We thank the reviewer for pointing out the inconsistency in Table 1. We would like to point out our numbers are only drawn from [9] and [44], not from the git repo or other implementations. The inconsistency comes from the fact that we drew 8-bit values from [44] which uses knowledge distillation (KD) and full precision from [9] which doesn’t. We have to disagree with the reviewer about the fairness of our comparison with AdderNet. The fact we used KD values in Table 1 actually obfuscates our contribution, because KD is a method known to improve accuracy, and we did not implement EuclidNets with KD. In fact, AdderNet 8-bit with KD is similar to EuclidNet 8-bit without KD, while EuclidNet if full precision is always better in terms of accuracy. To make the table more fair we should have not used KD values, since training EuclidNets with KD would also improve our accuracy. We would also like to add AdderNets is extremely slow to train with KD (a day for CIFAR10) and it simply does not train for 8-bit without KD. EuclidNet can be used out of the box since it takes advantage of being rewritten as a sum of convolutions. Even though we developed the homotopy method, it is not necessary to achieve high accuracy. Finally, we would like to add that AdderNet is still the most efficient architecture of the two, and they both can co-exist with their respective trade-offs.
>
> We hope to have addressed your points and shown how a short revision could improve our work, and let us know if you require any clarifications or further discussion.
>
> * AdderNet and its Minimalist Hardware Design for Energy-Efficient Artificial Intelligence, Yunhe Wang, Mingqiang Huang, Kai Han, Hanting Chen, Wei Zhang, Chunjing Xu, Dacheng Tao, 2021

---

> > ### Author Response · Authors · 2021-08-31
> > **Thank you for the high rating**
> >
> > Thank you for the high rating.  We hope that our response to the reviewers has helped.

---

### Official Review · Reviewer_yKQG · 2021-07-19

**Rating:** 5
**Confidence:** 4

**Summary:**

The authors propose to replace the traditional convolution operation with squared differences in vision models. Compared to convolution, squared differences can be implemented with fewer logic gates, thus increasing energy efficiency and potentially improving the performance of inference.

**Limitations And Societal Impact:**

Yes

**Main Review:**

The paper presents a new category of vision model, namely EuclidNets, which uses squared difference as the computational kernel. The intuition behind Euclidean nets is clear, since any multiplicative operation can be expressed in terms of Euclidean operations only. The results of the empirical evaluation are also promising, as EuclidNets can achieve similar accuracy as ordinary convolutional neural networks. The idea is sound to me. I believe this paper clearly articulates a contribution to the literature. However, the paper should be improved in the following aspects:
1) The organization and writing of the paper needs to be improved. For example, Table 1 is an important table containing the main results of EuclidNets and a comparison with the baseline and related work. However, the experimental setting is not presented, which makes it difficult for the audience to know whether the figures in the table are directly comparable. I suggest to move the paper to the experimental section with more detailed explanation.
2) Theoretical gate count estimation. I appreciate the theoretical gate count estimation in the paper and appendix. However, in terms of the actual circuit implementation, the logical synthesis tool will optimize the circuit, which may change the results. I suggest to obtain the gate count using either commercial or open source logical tools. In addition to the number of gates, the authors can also report the circuit depth or delay of the critical path in convolution, addition, and squared difference circuits.
3) Fair comparison with AdderNet. Both Table 1, 3, 4 contain results of AdderNet with and without knowledge distillation. However, the numbers reported in this paper do not match with numbers reported in AdderNet paper and github repo [1].  For example, the github repo of AdderNet claims 92.96% and 68.8% top-1 accuracy with knowledge distillation on ResNet-20 for CIFAR-10 and ResNet-18 for ImageNet, respectively. I think it should be more prudent to specify whether the results are obtained by reference to a paper, some open source implementation, or self-implementation. More importantly, the current comparison with AdderNet is not fair, because addition requires much fewer logic gates than squared differences. The authors should come up with a better way to compare the two and thus demonstrate the advantages of EuclidNets.

[1] https://github.com/huawei-noah/AdderNet/blob/e69b13dff4f64f859ff1d32f6d2ec2ecae2a2c1b/README.md

Nitpicks:
1) Title of Figure 4 ResNet 18 - Cifar-10
3) Line 172: Fine-tuning method in (7) is inspired by continuation methods in partial differential equations
2) Line 219: Training a quantized Seuclid is very similar similar to convolution


**Time Spent Reviewing:**

4

---

> ### Author Response · Authors · 2021-08-09
> **Response to comments**
>
> Thank you for taking the time to review our paper. It appears you have captured the main point of our paper which was improving efficiency while retaining accuracy close to convolutional neural networks. Also, thank you for recognizing the paper would be a contribution to the literature.
>
> However, you express valid concerns we would like to address. We believe points 1 and 3 could easily be addressed in a revision to the paper, while point 2 may be more appropriate for follow-up work. We go into detail next:
>
> 1. Regarding the missing experimental setting in Table 1, there are two different things to clarify: 1) we can easily add references to the rows of S_adder and BNN as the values were copied from references [9] and [44] already cited. We may also add a short paragraph describing the experimental setting as they appear in this reference. As for the remaining rows, we can add a sentence referring the reader to Section 6 where the experimental details are described. To address your more general issue of paper organization, the reason we chose to display Table 1 at the beginning is so our contributions and impact are clear from the start without overwhelming the reader with details. We still have more in depth justifications, but in the following sections of the paper. We hope being more careful in guiding the reader to those sections helps with your concern.
> 2. This is a great suggestion, but also a major additional step. Our understanding is that new architectures are already a major big step with demonstration of training and accuracy. The reason we provided an intuitive theoretical gate estimation is already to improve upon other literature papers that do not consider hardware (AdderNet works for example [9,44]). We may not have implemented hardware with the expertise of ShiftAddNet [46], because we focused instead in speeding up training by developing the homotopy method, which is not usually addressed in other papers. Additionally, only recently has a hardware implementation been done for AdderNet [*].
> 3. We thank the reviewer for pointing out the inconsistency in Table 1. We would like to point out our numbers are only drawn from [9] and [44], not from the git repo or other implementations. The inconsistency comes from the fact that we drew 8-bit values from [44] which uses knowledge distillation (KD) and full precision from [9] which doesn’t. We have to disagree with the reviewer about the fairness of our comparison with AdderNet. The fact we used KD values in Table 1 actually obfuscates our contribution, because KD is a method known to improve accuracy, and we did not implement EuclidNets with KD. In fact, AdderNet 8-bit with KD is similar to EuclidNet 8-bit without KD, while EuclidNet if full precision is always better in terms of accuracy. To make the table more fair we should have not used KD values, since training EuclidNets with KD would also improve our accuracy. We would also like to add AdderNets is extremely slow to train with KD (a day for CIFAR10) and it simply does not train for 8-bit without KD. EuclidNet can be used out of the box since it takes advantage of being rewritten as a sum of convolutions. Even though we developed the homotopy method, it is not necessary to achieve high accuracy. Finally, we would like to add that AdderNet is still the most efficient architecture of the two, and they both can co-exist with their respective trade-offs.
>
> We hope to have addressed your points and shown how a short revision could improve our work, and let us know if you require any clarifications or further discussion.
>
>
> [*] AdderNet and its Minimalist Hardware Design for Energy-Efficient Artificial Intelligence, Yunhe Wang, Mingqiang Huang, Kai Han, Hanting Chen, Wei Zhang, Chunjing Xu, Dacheng Tao, 2021

---

> > ### Author Response · Authors · 2021-08-31
> > **We have responded to your review**
> >
> > Thanks for pointing out that the idea is sound, that the intuition is clear, and that the results are promising.
> >
> > You gave three points of concern, and we gave a full response to all of them.  In fact, two of the three can be satisfied in a revision.   The middle point is more appropriate for a second paper, as was done in reference [*] above.
> >
> > In light of our response, we feel that the rating of 4 is overly harsh.
> > Currently two of the reviewers have given ratings of 7 and 6, and we hope that you can increase your rating to a 6 as well.

---

> > > ### Comment · Reviewer_yKQG · 2021-09-01
> > > **Response to the author's rebuttal**
> > >
> > > Thanks for the response. I am satisfied with the answers to questions 1 and 3. However, I am still not convinced that how much improvement EuclidNet can bring, since there is no hardware implementation. The authors provide gate count estimates for three different kernels, but do not provide an overall gate count estimate for the entire accelerator design. It is worth mentioning that the matrix multiplication unit in the hardware consumes only a small fraction of the area (about 30% in the TPU [1] and [2]). As a result, EuclidNet may only slightly reduce the complexity (in terms of area). I will raise my score to a 5 because the author addressed my other concerns.
> > >
> > > [1] In-Datacenter Performance Analysis of a Tensor Processing Unit, ISCA'17
> > >
> > > [2] Ten Lessons From Three Generations Shaped Google’s TPUv4i, ISCA'21

---

### Official Review · Reviewer_vThL · 2021-07-20

**Rating:** 6
**Confidence:** 2

**Summary:**

EuclidNets propose a new multiplication free method to conduct convolution in neural networks, which only involves subtraction and squaring, and achieves improved accuracy as compared with other baseline multiplication free methods while not sacrificing too much of the hardware efficiency, in terms of the number of logic gates used.

**Limitations And Societal Impact:**

It's addressed

**Main Review:**

Strength:
Overall this paper is
1. Clear and illustrative; provides the good reasoning and intuitive demonstration of why the proposed method is effective
2. propose a new and interesting direction which still leaves much room for future work if more hardware flexibility is considered

Weakness:
However, there are a few concerns which would be great if get addressed:
1. The accuracy improvements seems to be marginal as compared with the baselines; it would be good if the authors could conduct more trials of the experiments and provide the variance of the converged accuracy
2. The evaluated network variety seems to be limited. Currently only ResNet series are considered. It would be more informative if more networks or even tasks are considered to show the methods general effectiveness.
3. The hardware cost, in terms of the logic gates, seems still to be at the same magnitude against the multiplication network, through theatrical calculation. End2end (if possible) latency/energy/area consumption of the proposed method against the multiplication method would be good.
4. More theoretical proof of the EuclidNets' superiority over the baselines such as AdderNet in terms of converged optima or rate would be better. Otherwise, the method seems a little too empirical.

**Time Spent Reviewing:**

2 hours

---

> ### Author Response · Authors · 2021-08-09
> **author response**
>
> Thank you for taking the time to review our paper and recognizing both originality and intuitive reasoning as the main strengths. In particular, we agree that this direction has the chance to lead to much follow-up work. That said, there are some concerns (points 2-4) related to the scope of our paper. We believe most of these concerns can be addressed either by quick additions to our paper, or they are simply too broad and as we see it should be instead a follow up paper. We addressed all of these in detail next.
>
> 1. This is a great point but unfortunately, due to the training time required it is not possible to conduct multiple trials. As for the improvements being marginal, EuclidNets are only expected to achieve comparable accuracy to ConvNets, and the advantage lies in their efficiency. When compared to AdderNets, they are better in full precision, and comparable in 8-bits because the paper [44] uses a more sophisticated approach, knowledge distillation (KD). For a fairer comparison, we should have trained EuclidNets with KD as well. Another possibility would have been to include the results from [46] where 8-bit accuracy drops almost 10% from the full precision baseline, but we are not confident in these values as [46] also shows unusually low accuracy in AdderNet full precision.
> 2. This is a very good point, as more variety of networks would be more informative. However, this represents significant extra work. Similar works such as AdderNet [9] and ShiftAddNet [46] evaluate on similar networks to ours, and that what we based our selection of models on. However, [9,46] do consider VGG-small on CIFAR10 and CIFAR100 and this is something we can add quickly to our paper, as these are smaller datasets.
> 3. This is another good point, but represents an additional project. One of the main architectures of the field, AdderNet [9] has not considered this until quite recently [*]. Our understanding is that developing an accurate new architecture, an efficient training mechanism and an theoretical intuition that justifies efficiency was already a significant contribution to the field. However, we do think our natural next step would be to work on understanding such details about hardware.
> 4. This would be very interesting to expand on but out of the scope of the paper.
>
> We hope to have addressed your concerns and that the scope and impact of our paper is clearer now.
>
>   AdderNet and its Minimalist Hardware Design for Energy-Efficient Artificial Intelligence, Yunhe Wang, Mingqiang Huang, Kai Han, Hanting Chen, Wei Zhang, Chunjing Xu, Dacheng Tao, 2021

---

> > ### Author Response · Authors · 2021-08-31
> > **Thank you for your review of the paper and increasing the rating from 5 to 6**
> >
> > Thank you for your review of the paper and increasing the rating from 5 to 6

---

### Official Review · Reviewer_2baF · 2021-07-20

**Rating:** 5
**Confidence:** 4

**Summary:**

This work proposes EuclidNet, which replaces the multiplication in DNN with squared differences with less logic gates.
It is proved to be as expressive as convolution networks and able to be trained in GPUs with at most 2x more time.
The experiments are conducted on ResNet@CIFAR-10 and ImageNet.

**Limitations And Societal Impact:**

The authors clearly addressed the limitations of their work in Section 7.1

**Main Review:**

> + Originality: Are the tasks or methods new? Is the work a novel combination of well-known techniques? (This can be valuable!) Is it clear how this work differs from previous contributions? Is related work adequately cited?

Replacing multiplication with squared differences is relatively new and the analysis on the expressivity of the EuclidNet in Section 4.1 and the discussion on the logic gate cost in Section 4.2 also help on this point.

> + Quality: Is the submission technically sound? Are claims well supported (e.g., by theoretical analysis or experimental results)? Are the methods used appropriate? Is this a complete piece of work or work in progress? Are the authors careful and honest about evaluating both the strengths and weaknesses of their work?

1. The comparison with AdderNet: It seems that EuclidNet cost more logic gates than AdderNet but is not always better than AdderNet in terms of accuracy. In Table 1, AdderNet with 8-bit quantization achieves higher accuracy than EuclidNet on ImageNet, i.e., does it mean EuclidNet will perform worse in bigger models@bigger datasets? There is no other comparison with AdderNet and its corresponding variants [1,2] in Section 6, it would be better to add them.
2. Limited tasks: Currently all the tasks are conducted in Image Classification tasks, if the the transferability of the network to other tasks is verified to be valid, it will make this a more solid work.
3. Confusing experiments settings: In line 196-197, "We consider two different weight initialization for EuclidNets. First, we initialize randomly and second, we initialize from weights197 pre-trained on a convolutional network." However, in Table 3, all initialization function for EuclidNets are "Conv", so not sure if there is a real fair comparison with standard Conv under the random initialization setting. If all the performance of EuclidNets in Section 6 needs a pre-trained standard CNN, it would be better to at least add an ablation study to fine-tune standard CNN with the same training cost.

> + Clarity: Is the submission clearly written? Is it well organized? (If not, please make constructive suggestions for improving its clarity.) Does it adequately inform the reader? (Note that a superbly written paper provides enough information for an expert reader to reproduce its results.)

The submission is clearly written and well organized.

For the reproducibility, the code seems to be not in the Appendix and code link the in submission seems to be invalid. It would be better if the authors can share their implementations in GPU to verify their claim about "EuclidNet training is 2x n the worst case and their implementation is natural in deep learning frameworks such as PyTorch and Tensorflow".

> + Significance: Are the results important? Are others (researchers or practitioners) likely to use the ideas or build on them? Does the submission address a difficult task in a better way than previous work? Does it advance the state of the art in a demonstrable way? Does it provide unique data, unique conclusions about existing data, or a unique theoretical or experimental approach?

Because the comparison with AdderNet is not well illustrated and analyzed, I think the significance is slightly lower than the average of NeurIPS accepted papers.

[1] Kernel Based Progressive Distillation for Adder Neural Networks. Yixing Xu, Chang Xu, Xinghao Chen, Wei Zhang, Chunjing XU, Yunhe Wang. NeurIPS, 2020.

[2] ShiftAddNet: A Hardware-Inspired Deep Network. Haoran You, Xiaohan Chen, Yongan Zhang, Chaojian Li, Sicheng Li, Zihao Liu, Zhangyang Wang, Yingyan Lin. NeurIPS, 2020.

**Time Spent Reviewing:**

3.5

---

> ### Author Response · Authors · 2021-08-09
> **response**
>
> Thank you for reviewing our paper and summarizing the main idea. Thank you for describing our paper as well written, organized and the originality of Section 4.
>
> As for the expressed concerns, we believe points 1, 3 and 4 can be easily addressed in a short rewrite, while point 2 is more relevant for future work. We attempt to address each of these in detail next.
>
> 1. The comparison with AdderNet: First, we would like to point out the difference in 8-bit accuracy is 0.21% on ImageNet. So EuclidNet performs similarly to AdderNet and not worse, as this difference is not large enough when stochasticity is involved. For example, we have a larger difference between our reported CNN accuracy and [9]’s CNN reported accuracy. However, the question remains valid. That said, we would like to clarify that in Table 1 the 8-bit AdderNet is retrieved from [44], where knowledge distillation (KD) is used. As far as we know, it is not possible to train AdderNet in 8-bit to full-precision accuracies without using this more sophisticated method ([2] has good plots of 8-bit AdderNet without KD). Hence, EuclidNet is able to achieve similar accuracy without KD and trains faster and easier by relying on optimized computations for multiplications and convolutional initialization. In this case, instead of having a trade-off between accuracy and efficiency, we have a trade-off between training and inference efficiency. Since this is a great point, we would update a future version of the paper to include this more in-depth and clearer comparison. As for comparison to [1], we believe it is not the fairest as using KD may also improve EuclidNet accuracy, although it is still valid to mention. As for [2], we were not able to use their code and the results in Table 2 are lower for AdderNet than those reported in [9], and subsequently ShiftAddNet always displays lower accuracy. However, we may add it as a row in a subsequent update to the paper.
> 2. This is a very good point, more tasks would lead to more solid work. However, this would be an additional project. For example, both [1] and [2] cited by the reviewer consider only image classification tasks, with [1] only studying CIFAR10 and CIFAR100. Some follow-up work to AdderNet does indicate we would be able to apply it to other tasks and these are citations we can easily add to our paper.
> 3. We are not sure we understood this question. It appears to us to be derived from poor formatting on Table 3 which leads to confusion. All initialization functions for EuclidNets are not "Conv", and instead one is “Random” with 450 epochs and the other is “Conv” with two different types of homotopy as expected. AS for $S_conv$ we have only one initialization possibility “Random” with 400 epochs. Please let us know if we understood your question correctly and it was indeed a formatting error.
> 4. We apologize for the invalid link and provide an updated link for our code soon. We would like to add that EuclidNets can be written as a sum of two convolutions and a fixed squared term so the speed would in the worst case be 3x as slow as ConvNets per training step. Running our code will show we observe a faster time, but not as slow as for example the current available implementation of AdderNet [1], which does not use CUDA.
>
> We agree with the reviewer that we can clarify some points in the AdderNet comparison as we discussed above, and hope this will strengthen our paper and make it more suitable for NeurIPS.

---

> > ### Author Response · Authors · 2021-08-14
> > **Code link**
> >
> > Please access the code here: https://drive.google.com/drive/folders/1hHZxR2FvAfX5o7uYJQAW5rP6SezMBBVu.

---

> ### Author Response · Authors · 2021-08-31
> **Awaiting a response**
>
> In our response we responded to your main concern, that the comparison AdderNet requires more detail.
>
> Since we have show fully how to perform this comparison, in particular, in our point 1 of our answer, we ask that you increase your rating from 5 to 6.

---

> > ### Author Response · Authors · 2021-09-02
> > **AdderNet Comparison - possible misunderstanding**
> >
> > We would like to add that Point 3 - which also argues the comparison is not fair - appears to come from a misunderstanding. We address this in our response below, so please take this into account when considering whether to raise our score.

---

### Author Response · Authors · 2021-08-14
**Sharing the code**

We would like to share an updated link to the code to support our submission, as requested: https://drive.google.com/drive/folders/1hHZxR2FvAfX5o7uYJQAW5rP6SezMBBVu.

---

### Author Response · Authors · 2021-08-31
**Thanks to Reviewer vThL for increasing the rating.   Please respond to our comments.**

Hi, we believe that we have satisfied the concerns of the referees.
Thanks to reviewer Reviewer vThL who has increased the rating from 5 to 6.  We kindly request that the other  reviewers reconsider their ratings, in light of our clarifications, posting, code, and comparison with AdderNet

---

### Decision · Program_Chairs · 2021-09-27

**Decision:**

Reject

**Comment:**

The discussion phase greatly helped clarify several concerns raised by reviewers around some experimental comparisons, particularly around the use of knowledge distillation in some of the experimental results reported from other papers. One sigficant outstanding issue that came up during the discussion phase is around the complexity of EuclidNets and whether gate counts are appropriate estimates of complexity or not. Specifically, one of the reviewers noted that the matrix multiplication unit in TPUs consumes only a small fraction (30%) of the area, thus reducing the impact that EuclidNets would have compared to that platform. Other reviewers noted that comparisons to sparse training methods (fp16, bp16) are necessary. Overall, the lack of measures of complexity other than gate counts makes assessing the contribution of the method particularly challenging, particularly given that hardware implementations of AdderNet, albeit recent, do exist.